# Effects of emerging SARS-CoV-2 on total and cause-specific maternal mortality: A natural experiment in Chile during the peak of the outbreak, 2020–2021

**Yordanis Enriquez**[1], **María Elena Critto**[2], **Ruth Weinberg**[3], **Lenin de Janon Quevedo**[2], **Aliro Galleguillos**[4], **Elard Koch**[5] *

**1** Facultad de Ciencias de la Salud, Universidad Católica Sedes Sapientiae, Lima, Peru, **2** Facultad de Ciencias Médicas, Pontificia Universidad Católica, Buenos Aires, Argentina, **3** Facultad de Medicina, Universidad Nacional de Buenos Aires, Buenos Aires, Argentina, **4** Facultad de Medicina, Universidad de Chile, Santiago de Chile, Chile, **5** MELISA Institute, Concepcion, Chile

* ekoch@melisainstitute.org

**Data Availability Statement:** The data underlying this article is available from Departamento de

## Abstract

This study estimated the effects of the COVID-19 pandemic on maternal mortality in Chile between 2020 and 2021. A natural experiment was conducted using official data on maternal deaths and live births (LBs) between 1997 and 2021. The effects of the SARS-CoV-2 outbreak were evaluated using interrupted time series (ITS) and an autoregressive integrated moving average (ARIMA) model to forecast the expected rates on MMR and 95% confidence intervals (95% CI). In Chile, following World Health Organization suggestions, maternal deaths aggravated by SARS-CoV-2 are assigned to code O98.5 (non-respiratory infectious indirect) accompanied by code U07.1 or U07.2, depending on confirmation of the presence or absence of the virus. ITS analysis revealed that the SARS-CoV-2 outbreak impacted the MMR due to indirect causes, with a greater increase in indirect nonrespiratory causes than respiratory causes. The ARIMA forecast was consistent with ITS, showing that the expected MMR for indirect causes (3.44 in 2020 and 1.55 in 2021) was substantially lower than the observed rates (9.65 in 2020 and 7.46/100.000 LBs in 2021). For nonrespiratory indirect causes, the observed values of the MMR for 2020 (8.77/100.000 LBs) and 2021 (7.46/100.000 LBs) were double the predicted values of 4.02 (95% CI: 0.44–7.61) and 3.83 (95% CI: -0.12–7.79), respectively. A lower effect was observed on direct obstetrical deaths. During 2020–2021, there was a rise in the MMR in Chile attributable to SARS-CoV-2. The pandemic contributed to an escalation in the MMR due to indirect causes, particularly nonrespiratory and infectious causes. MMR due to direct obstetric causes were less affected. This suggests that the pandemic disproportionately affected maternal health by exacerbating conditions unrelated to pregnancy, childbirth, or postpartum, more than those directly linked to obstetric complications.

Estadísticas de Información de Salud del Gobierno de Chile, https://deis.minsal.cl/. The derived data generated in this research is available on supplementary material.

**Funding:** This study was supported by research grants EPI-092018-01 and ONE-052021-01, both granted by FISAR http://www.fisarchile.org/. The funders had no role in study design, data collection and analysis, decision to publish, or preparation of the manuscript.

**Competing interests:** The authors have declared that no competing interests exist.

## Introduction

At the end of January 2020, the World Health Organization (WHO) declared an outbreak of a new coronavirus (SARS-CoV-2), classifying it as a public health emergency of international importance and later a pandemic [1]. The responses to the pandemic varied greatly among countries and were influenced by factors such as available resources, protection and prevention supplies, training and preparation of health personnel, data monitoring, health system integration, availability of critical beds and mechanical ventilators, and accessibility to medical services [2].

In addition, despite the measures taken to prioritize access to health in the population in the face of SARS-CoV-2, a decrease in the use of medical services during the pandemic was observed in some middle- and low-income countries [3]. The increase in mortality from SARS-CoV-2 unevenly affected vulnerable populations, such as elderly people living in nursing homes and people with preexisting diseases and comorbidities [4]. The statistical data of each country and the scientific analysis thereof allow us to observe how the COVID-19 pandemic has had direct and indirect impacts on the health of the population worldwide, on access to services, and, in particular, on mortality [3, 5]. In this sense, the data from the mortality registers and their analysis through natural experiments allow us to establish the magnitude of the specific impact that the pandemic has had on total and cause-specific mortality.

The pandemic has notably impacted preventable deaths, including maternal deaths, thereby interrupting the progress observed in recent decades in reducing maternal mortality (MM) worldwide [6]. In light of this, the WHO responded by issuing an epidemiological alert. This alert highlighted a higher risk of severe clinical manifestations of SARS-CoV-2 among pregnant women [7].

Responding to the same crisis, in February 2020, the Ministry of Health of Chile decreed a Health Alert for the whole country to take measures to deal with the epidemic in its early stage [8]. Despite having a relatively low maternal mortality ratios (MMR: 19.1/100,000 LBs in 2019) compared to other countries in the region, at the national level, the excess of MM from causes that could be attributable to SARS-CoV-2 has not yet been estimated. Such an estimate would provide not only a measure of the MM attributable to the virus but also of the effect it had on maternal mortality from specific causes [9]. In addition, the comparison between maternal mortality data prior to the pandemic period and those attributable to the pandemic makes it possible to clarify the relevance of the impact on diverse groups of MM causes. This information is crucial for the development and execution of strategic programs and policies aimed at identifying and mitigating preventable diseases and deaths. Given this scenario, our study proposed to assess the impact of the COVID-19 pandemic on the total and cause-specific MMR during its first two years in Chile.

## Material and methods

### Study design and data sources

This was a study with an observational and ecological design using official maternal mortality data at the national level.

Time series of population data on maternal deaths from 1997 to 2021 published in database format by the Department of Health Statistics and Information (DEIS) of the Ministry of Health of Chile were prepared [10]. The death records for 2020 and 2021 corresponded to provisional data updated by this organization on May 17, 2022. However, corrected live births came from the National Institute of Statistics of the same country, published in the Vital Statistics Yearbooks [11]. The calculated maternal mortality ratios (MMRs) are available on supplementary material (S1 Data).

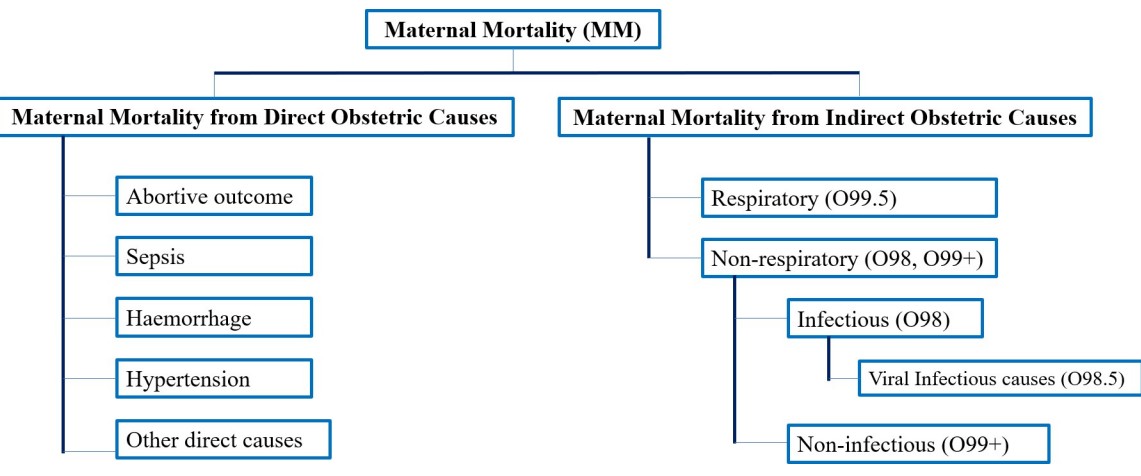

**Fig 1. Flowchart of maternal mortality classified by cause groups.**

## Study variables

The total MMR was calculated as the ratio of maternal deaths (ICD-10: Codes O00 to O99, excluding codes O96 and O97) to the number of corrected live births, multiplied by 100,000. The MMR was categorized into groups of direct and indirect obstetric causes (Fig 1).

The MMR due to direct obstetric causes referred to maternal deaths resulting from obstetric complications during pregnancy, childbirth, and the postpartum period, as well as from interventions during these stages (ICD-10: Codes O00 to O99, excluding O98 and O99) [11, 12]. The rate was calculated per 100,000 live births.

The MMR due to indirect obstetric causes included deaths that resulted from a preexisting disease or one that developed during pregnancy and was exacerbated by the physiological effects of pregnancy [13]. This category of maternal deaths (ICD-10: Codes O98 and O99) per 100,000 live births was further divided into indirect respiratory and nonrespiratory obstetric causes.

The indirect respiratory obstetric MMR included maternal deaths caused by diseases of the respiratory system that were exacerbated during pregnancy, childbirth, and puerperium (ICD-10: Code O99.5). The indirect nonrespiratory obstetric MMR comprised all deaths from indirect causes, except for respiratory complications. Within this subgroup, it was possible to identify indirect infectious causes (ICD-10: O98) and indirect noninfectious causes (ICD-10: O99, except O99.5).

In Chile, following WHO suggestions, maternal deaths aggravated by SARS-CoV-2 were assigned to code O98.5 (nonrespiratory infectious indirect) as the primary cause of death when secondary causes included codes U07.1 orU07.2, depending on confirmation of the presence or absence of the virus [12, 13].

For the subgroups of respiratory and infectious causes of maternal mortality, due to the low frequency of events, the maternal mortality rate was amplified by 1,000,000 LBs. For the years in which no cases were reported, one case per 1,000,000 LBs was attributed.

## Selection criteria

The study focused on the main cause of maternal death as published by DEIS Chile; secondary or underlying causes were not published and, thus, were excluded. This approach is consistent with DEIS Chile's method of reporting the primary cause of death. For maternal deaths related to SARS-CoV-2, code O98.5 was assigned as the principal cause.

## Statistical analysis

The effect of the SARS-CoV-2 on different maternal mortality groups was first assessed using an interrupted time-series (ITS) approach in parallel trends in cause-specific mortality.

Three coefficients were estimated in this model: $\beta_0$ represents a constant or starting point for the variable of interest; $\beta_1$ is the change in outcome associated with an increase in the unit of time $T_t$, which is interpreted as the preintervention trend; $\beta_2$ is the level of immediate change after the $X_t$ intervention, which is specified as a dummy variable (preintervention period = 0 and postintervention period = 1); and $\beta_3$ indicates the change in slope after the intervention with respect to the initial trend.

The ITS model considered the occurrence of a natural event such as the COVID-19 pandemic and assessed significant immediate trends and changes or sizes of effects attributable to the event of interest using the linear minimum squares method. Coefficients with standard Newey–West errors were estimated to evaluate autocorrelation and possible heteroscedasticity. This robust approach adjusts standard errors and accounts for potential serial correlation in the time series data. Additionally, we explored the Prais-Winsten method. We conducted sensitivity analyses to assess the robustness of our results varying autocorrelation structures of the series.

The effect of the SARS-CoV-2 on the MMR was then estimated with an autoregressive integrated moving average (ARIMA) model [14]. First, the stationarity of the time series was analysed using the Dickey-Fuller test. The use of the ARIMA model avoided overestimating the statistical significance of the effect of the intervention produced by autocorrelation. The effect of SARS-CoV-2 on the variables of interest was evaluated with a forecast for the years 2020 and 2021 as if this had not been verified in 2020, thus calculating an estimate of the expected MMR in the absence of the pandemic for those two years. First, for the series, a one-step-ahead training set was determined for the years 2015–2019. Then, the forecasts were compared with observed annual MMR values, identifying a difference in mortality attributable to the pandemic. For the MMR due to respiratory causes, no prediction was made for 2021, as no maternal deaths were reported in this group for this year.

Subsequently, to evaluate the accuracy of the prediction of the different models, the mean absolute error (MAE) was determined. The MAE depends on the prediction measurement scale and is therefore expressed in the same unit of measure as the analysed data, giving the size of the prediction error with an average of the difference between prediction and observation. In addition, the mean absolute percent error of prediction (MAPE) was estimated. This expresses the percentage of error attributable to the prediction and does not depend on the measurement scale of the observed data. However, this metric takes on an undefined value when zero values exist in the analysed series. The following categorization was assumed to determine the predictive ability of the models: MAPE <10 (highly accurate prediction), 10–20 (good prediction), 20–50 (reasonable prediction), and >50 (inaccurate prediction). Likewise, the percentage accuracy of the predictions was calculated by subtracting the magnitude of the error expressed in the MAPE from 100 to express the variability explained by the model. In these analyses, the software Stata version 15.1 (Stata Corp, College Station) was used, and the corresponding 95% confidence intervals (95% CIs) were estimated for the measurements.

## Ethical approval

This was a secondary data analysis of publicly available information. The source of the information is the official anonymous maternal mortality records of Chile published by the DEIS (available at https://deis.minsal.cl/). The study protocol was approved by the Ethics Committee of the Catholic University Sedes Sapientiae, Lima, Peru.

**Table 1. Effect of the SARS-CoV-2 in 2020 on the total maternal mortality ratio (MMR) and by groups of mortality groups of causes, in Chile, 1997–2021.** Interrupted time series analysis.

| | Initial slope | | | Change in level | | | Slope change | | | Segment 2020–2021 | | |
|---|---|---|---|---|---|---|---|---|---|---|---|---|
| | β1 | 95% CI | p value | β2 | 95% CI | p-value | β3 | 95% CI | p value | β4 _ | 95% CI | p value |
| MMR total | -0.39 | -0.53–0.24 | <0.000 | 6.44 | 4.94–7.93 | <0.000 | -3.11 | -3.26- -2.97 | <0.000 | -3.51 | -3.51- -3.52 | <0.000 |
| MMR by groups of causes | | | | | | | | | | | | |
| Direct causes | -0.43 | -0.62–0.23 | <0.000 | 2.12 | 0.14–4.11 | 0.037 | -0.88 | -1.08- -0.68 | <0.000 | -1.31 | -1.31 - -1.32 | <0.000 |
| Indirect causes | 0.02 | -0.10–0.16 | 0.66 | 4.36 | 2.48–6.24 | <0.000 | -2.22 | -2.35- -2.08 | <0.000 | -2.19 | - | - |
| Indirect respiratory causes | -0.01 | -0.03–0.00 | 0.15 | 0.76 | 0.51–1.01 | <0.000 | -0.85 | -0.87- -0.83 | <0.000 | -0.86 | -0.87- -0.86 | <0.000 |
| Indirect non-respiratory causes | 0.04 | -0.07–0.16 | 0.24 | 3.59 | 1.91–5.27 | <0.000 | -1.35 | -1.47- -1.23 | <0.000 | -1.31 | - | - |
| Indirect infectious causes | -0.02 | -0.05–0.00 | 0.06 | 2.93 | 2.63–3.23 | <0.000 | 2.22 | 2.19–2.25 | <0.000 | 2.19 | 2.19–2.20 | <0.000 |
| Indirect non-infectious causes | 0.69 | -0.05–0.19 | 0.27 | 0.65 | -1.00–2.31 | 0.42 | -3.58 | -3.70–3.45 | <0.000 | -3.51 | 3.51–3.52 | <0.000 |

## Results

### Interrupted time series analysis

The effect of SARS-CoV-2 in 2020 on the diverse groups of the MMR is detailed in Table 1 and plotted in Fig 2. Thus, the interrupted time series (ITS) analysis revealed that the total MMR had a significant level change in 2020, corresponding to 6.44 (95% CI: 4.94–7.93) deaths per 100,000 LBs. However, in the period immediately after, this trend showed a significant decreasing change of 3.11 deaths per 100,000 LBs (95% CI: -3.26- -2.97) (Fig 2A).

MMR: Maternal Mortality Ratio per 100,00 live birth; $\beta_1$: pending before the intervention; $\beta_2$: level change in the period immediately after the start of the intervention; $\beta_3$: interaction term between the time from the start of the study and the variable that represents the pre- and post-intervention periods, indicating an effect over time; $\beta_4$: calculated trend of the model through the linear combination of coefficients ($\beta_1 + \beta_3$).

Regarding the direct-cause MMRs, there was a significant level change in 2020, corresponding to 2.12 (95% CI: 0.14–4.11) deaths per 100,000 LBs. However, in the period immediately after, this trend showed a decreasing change of 0.88 deaths per 100,000 LBs (95% CI: -1.08- -0.68) (Fig 2B). Likewise, the MMR due to indirect obstetric causes reported a significant level change of 4.36 deaths per 100,000 LBs (95% CI: 2.48–6.24) in 2020 (Fig 2C). Then, in the period immediately after, the trend decreased by 2.22 deaths per 100,000 LBs (95% CI: -2.35- -2.08).

Within the indirect MMR, the group of indirect nonrespiratory causes showed a greater level change in 2020 (Fig 2E) compared to the group of indirect respiratory causes (Fig 2D) (3.59 per 100,000 LBs vs. 0.76 per 100,000, respectively). However, in the period after 2020, both groups of indirect maternal deaths showed a significant decrease in the trend, being of greater magnitude in the group of nonrespiratory causes (1.35 per 100,000 LBs vs. 0.85 per 100,000, respectively).

On the other hand, the MMR due to indirect infectious causes reported statistically significant changes in the trend, with a significant change in level in 2020 of 2.93 deaths per 100,000 LBs (95% CI: 2.63–3.23), followed by an annual increase of 2.22 deaths per 100,000 LBs (95% CI: 2.19–2.25), in the period immediately after the onset of the pandemic (Fig 2F). In a similar vein, the MMR due to indirect noninfectious causes showed a significant decreasing change only in the period after 2020, with 3.58 deaths per 100,000 LBs (95% CI: -3.70–3.45) (Fig 2G).

### Forecast of the MMR with ARIMA models

The graphical inspection of the time series indicated the presence of trends in all groups of MMRs. In this way, the seasonality and subsequent differentiation of the time series of the

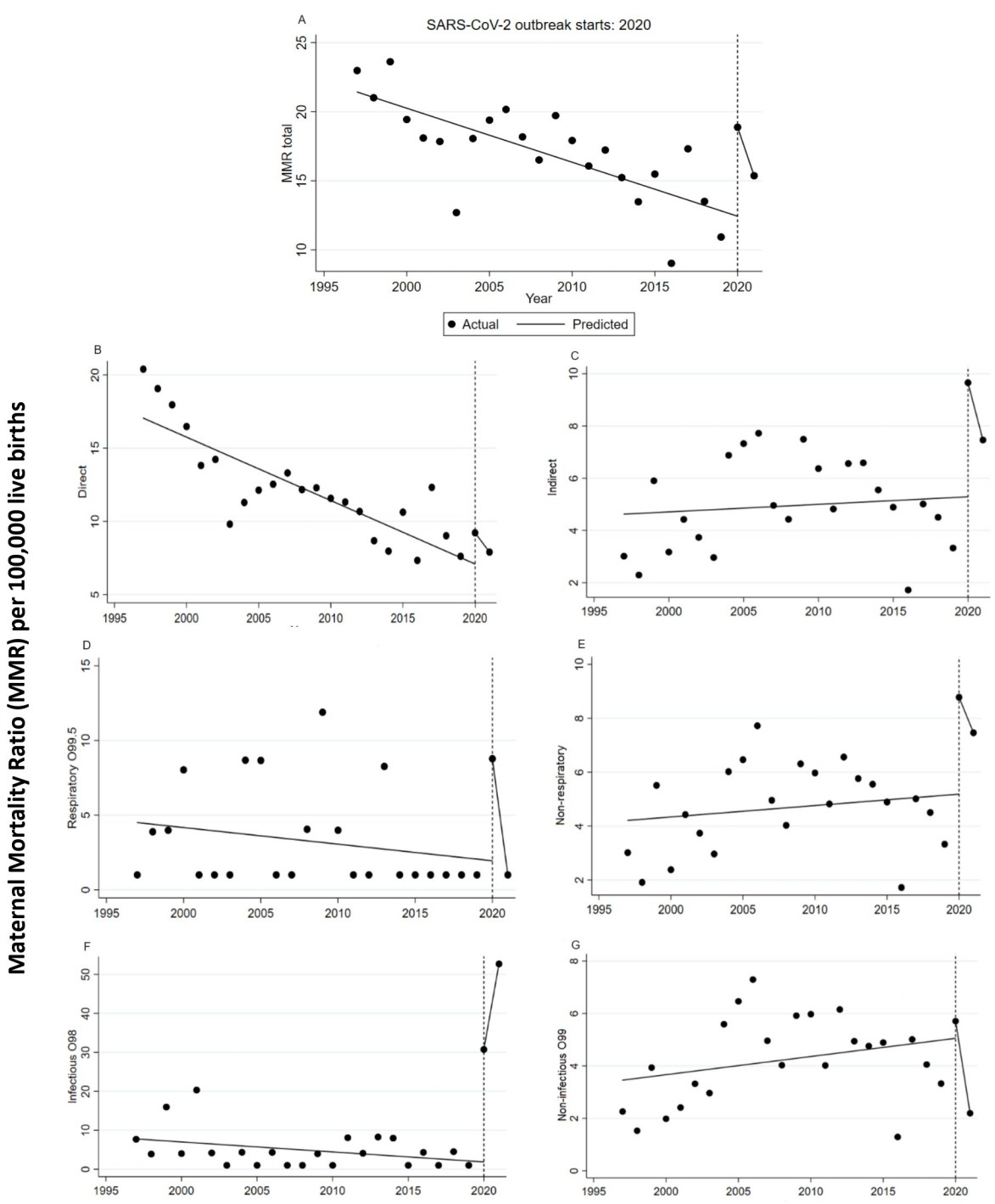

**Fig 2.** Interrupted time series analyses used to evaluate the effect of the 2020 SARS-CoV-2 on the total maternal mortality ratio (A) and maternal mortality ratios due to total direct causes (B), total indirect causes (C), respiratory causes (D), non-respiratory causes (E), infectious causes (F), non-infectious causes (G), using official records from 1997–2021. Vertical lines represent the year for the emergence of the SARS-CoV-2 influenza virus. A significant change in level due to the pandemic was identified for a, c, d, e, f, and g. However, there was no evidence of a slope change for b. An increase was observed in the following year for f, suggesting that the outbreak had an impact on mortality due to indirect causes.

**Table 2. Impact of the SARS-CoV-2 on the total maternal mortality ratio (MMR) and by groups of causes of mortality, in Chile (1997–2021).** Observed values versus prediction without pandemic. Forecast analysis with ARIMA models.

| | Year | Observed | Prediction | 95% CI | Delta | (p, d, q) [a] | MAPE [b] |
|---|---|---|---|---|---|---|---|
| MMR Total | 2020 | 18.88 | 15.28 | 8.47–22.08 | 3.60 | (2, 2, 0) | 15.19 |
| | 2021 | 15.37 | 10.73 | 4.45–19.84 | 4.64 | | |
| | 2022 | - | 11.73 | 5.76–23.99 | | | |
| RMM by groups of causes | | | | | | | |
| Direct causes | 2020 | 9.22 | 7.58 | 3.71–11.46 | 1.64 | (1, 2, 1) | 12.20 |
| | 2021 | 7.90 | 6.90 | 2.32–11.52 | 1.00 | | |
| Indirect causes | 2020 | 9.65 | 3.44 | -0.13–7.01 | 6.21 | (4, 2, 0) | 27.27 |
| | 2021 | 7.46 | 1.55 | -3.24–6.34 | 5.91 | | |
| Indirect respiratory causes* | 2020 | 8.80 | 1.00 | -4.30–6.30 | 7.80 | (3, 1, 0) | 109.33 |
| | 2021 | 0.00 | | | | | |
| Indirect no respiratory causes | 2020 | 8.77 | 4.02 | 0.44–7.61 | 4.75 | (2, 2, 1) | 34.63 |
| | 2021 | 7.46 | 3.83 | -0.12–7.79 | 3.63 | | |
| Indirect infectious causes* | 2020 | 30.70 | 3.45 | -6.15–13.06 | 27.25 | (1, 1, 0) | 154.51 |
| | 2021 | 52.61 | 1.73 | -8.29–11.76 | 50.88 | | |
| Indirect non-infectious causes | 2020 | 5.70 | 3.23 | -0.02–6.5 | 2.47 | (0, 2, 2) | 35.16 |
| | 2021 | 2.19 | 2.79 | -0.64–6.6 | -0.60 | | |

95% CI, 95% confidence interval

[a] Model components, where p represents the largest number of lags of the autoregressive parameter, d is the degree of differentiation of the series, and q is the largest number of components of the moving average

[b] MAPE: Mean Absolute Percent Error of prediction.

*Maternal mortality ratio (MMR) per 1,000,000 live births.

mortality groups were verified. Supplementary table (S1 Table) reports the parameters considered for the assessment of the models selected for the forecasts. For this purpose, the highest logarithm of likelihood, the lowest Akaike information criterion (AIC), and the lowest Bayesian information criterion (BIC) were considered for each model.

Based on the univariate ARIMA analysis, adjusted for the preinstallation data of the SARS-CoV-2, a forecast of expected cases was developed for 2020 and 2021. The observed and predicted values are detailed in Table 2 and plotted in Fig 3. The prediction reflected the MMR that might have been reported if the pandemic had not occurred. In 2020 and 2021, MMRs of 18.88 and 15.37 per 100,000 LBs were reported, respectively. The prediction for the first year showed 15.28 maternal deaths per 100,000 live births (95% CI: 8.47–22.08) and 10.73 (95% CI: 4.45–19.84) for 2021 (Fig 3A). Consequently, for the first year (2020), the maternal mortality difference attributable to the pandemic was 3.60 deaths per 100,000 live births. Among the possible candidates, the best model identified for the total MMR had the components (2,2,0). The accuracy of this prediction was 87.69%, with an MAE of 2.44 maternal deaths per 100,000 LBs. However, for 2021, the difference attributable to the pandemic was 4.64 deaths per 100,000 LBs (see Table 2).

On the other hand, for 2020 and 2021, the direct-cause MMRs were 9.22 and 7.90 per 100,000 LBs, respectively. In 2020, the ARIMA model prediction of maternal deaths per 100,000 live births for this group was 7.58 (95% CI: 3.71–11.46) and 6.90 (95% CI: 2.32–11.52) for 2021 (Fig 3B). Thus, the difference in maternal mortality attributable to the pandemic for 2020 was 1.64 deaths per 100,000 live births. Also, for 2021, the predicted value for this group of causes (6.90 per 100,000 LBs) was lower than the observed (see Table 2). The prediction accuracy for direct causes was 79.67%, and the model had the components (1,2,1), with an MAE of 1.30 maternal deaths per 100,000 LBs.

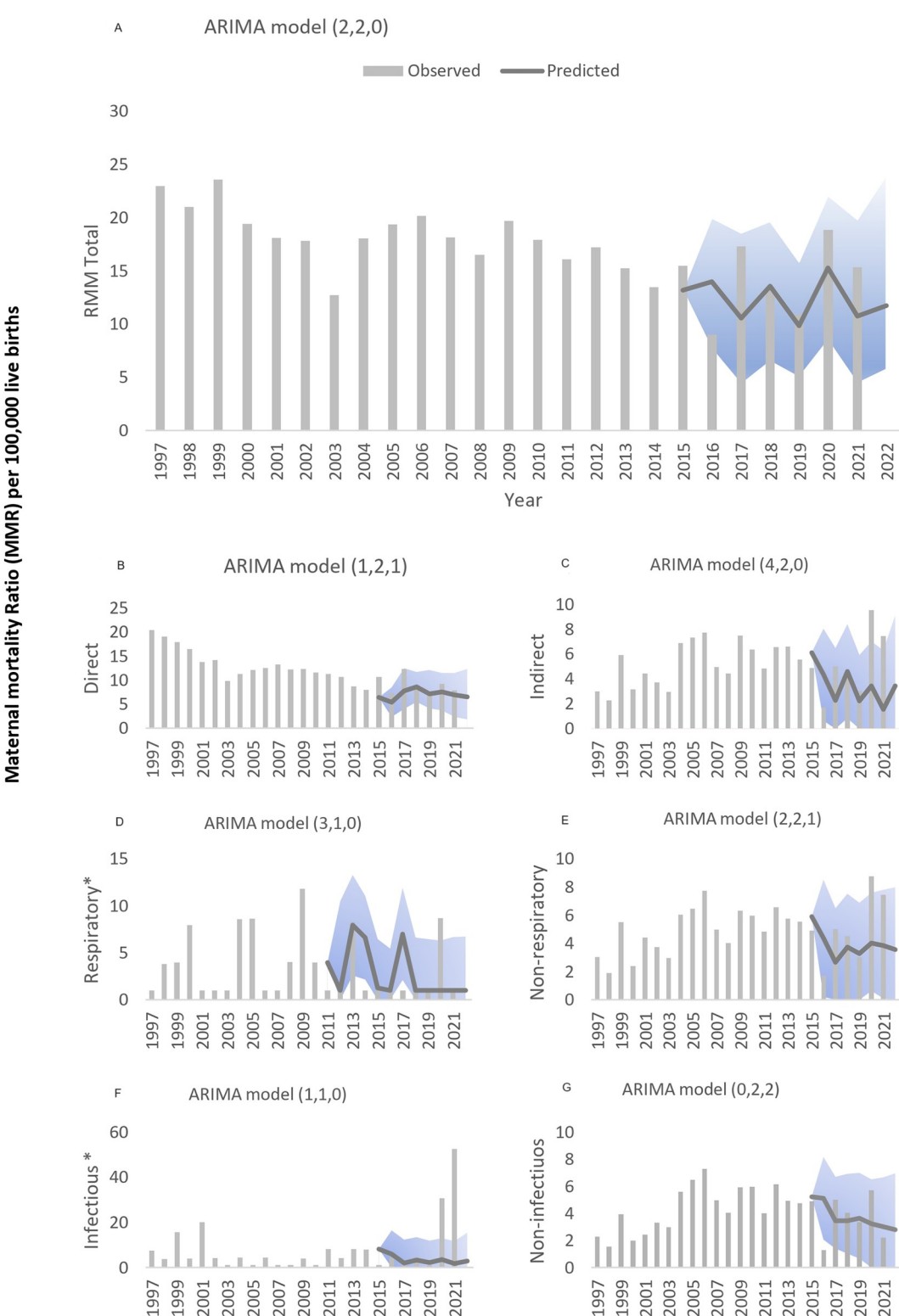

**Fig 3.** Expected maternal mortality ratio vs observed during the COVID-19 pandemic in Chile for (A) total maternal mortality ratio, (B) maternal mortality ratios due to total direct causes, (C) total indirect causes, (D) respiratory causes, (E) non-respiratory causes, (F) infectious causes, (G) non-infectious causes. * Maternal mortality ratio (MMR) per 1,000,000 live births. The line represents predicted values of the ARIMA models of MMR in Chile for total and specifics causes of maternal mortality ratio. Bars represents historical data. Light grey represents confidence intervals (CI 95%) for predicted MMR based on historical trends.

The best model for the MMR due to indirect causes was (4,2,0). The MMRs in this group were 9.65 and 7.46 per 100,000 LBs for 2020 and 2021, respectively. In contrast, the ARIMA model predictions for the same period were 3.44 (95% CI: -0.13–7.01) and 1.55 (95% CI: -3.24–6.34) (Fig 3C). In the comparison between groups of direct and indirect causes of death, this group had the greatest difference attributable to the pandemic, with an MMR of 6.21 per 100,000 LBs in 2020 and 5.91 in 2021, with an accuracy of 72.73% and an MAE of 1.16 maternal deaths per 100,000 LBs.

Regarding the group of indirect nonrespiratory causes, the observed values of maternal mortality in 2020 (MMR 8.77/100,000 LBs) and in 2021 (MMR 7.46/100,000 LBs) almost doubled the prediction. In this regard, the forecasts in the absence of a pandemic were 4.02 (95% CI: 0.44–7.61) and 3.83 deaths per 100,000 LBs, respectively (Fig 3E). The difference in maternal mortality attributable to the SARS-CoV-2 would be an MMR of 4.75 (2020) and 3.63 maternal deaths per 100,000 LBs in 2021. The best model for this group of MMRs was (2,2,1), while the ARIMA model prediction for the time series had an accuracy of 65.37%.

Finally, in the group of indirect infectious causes of mortality, the MMR was 30.70/ 1,000,000 LBs in 2020, with an increase in the following year reaching 52.61/1,000,000 LBs. The prediction for 2020 was 3.45 (95% CI: -6.15–13.06) maternal deaths/1,000,000 LBs, and for 2021, it was 1.73 (95% CI: -8.29–11.76) maternal deaths/1,000,000 LBs (Fig 3F). Thus, the increase in mortality attributable to the pandemic would be 27.25 maternal deaths/1,000,000 LBs in 2020 and 50.88 maternal deaths/1,000,000 LBs in 2021. In the prediction, the best model for this series had the parameters (1,1,0) and an MAE of 3.22 maternal deaths per 1,000,000 LBs. The information for the other groups evaluated is shown in Table 2.

## Discussion

In order to identify the excess maternal mortality associated with the SARS-CoV-2, this population-based study used ARIMA predictive models to estimate expected death rates in the absence of the pandemic event. Using this analytic approach, an increase in the observed total MMR for 2020–2021 relative to the expected values was confirmed. The finding was consistent with the results of an interrupted time series analysis for the secular trend. The latter showed that SARS-CoV-2 had an increasing effect on maternal mortality in Chile. There was a specific increase in the MMR due to indirect causes, specifically due to non-respiratory, infectious, and respiratory causes. Furthermore, there was an uptick in MMR attributed to direct obstetric causes between 2020 and 2021. However, this increase was comparatively smaller than the rise observed in indirect causes of death.

The findings in Chile confirm preliminary surveillance reports during the outbreak. A Mexican study reported an increase in the MMR of 11.3/100,000 LBs in 2020 (42.4/100,000 LBs in 2020 compared to an MMR of 31.1/100,000 LBs in 2019) [15]. Other studies reported an increase in maternal mortality associated with the pandemic in Brazil [16, 17] and Colombia [6]. In this regard, some factors that would explain this increase could be difficulties in accessing maternal health care and controls, as well as interruptions in specialized services, mobility restrictions, and other diseases that were not treated in time, particularly for high-risk pregnancies [18–21]. Likewise, the role of the fewer resources allocated to health, together with economic inequalities, may have increased the risk of maternal death from SARS-CoV-2 [17]. It is expected that difficulties in accessing health care among pregnant women would have an impact on their health and, therefore, on mortality from direct and indirect causes [22]. In this sense, from 2020–2021, an excess of maternal mortality from direct obstetric causes associated with the pandemic was also observed in Chile. These findings contrast with the results of a natural experiment conducted in Argentina with maternal mortality data (1980–2017) [14]. The

study found no effect of the H1N1 pandemic on the MMR due to direct causes such as hypertension, haemorrhage, abortion outcomes, or other direct obstetric causes.

An uptick in the MMR due to direct causes was observed in 2021. Coincidentally, ITS analysis showed an effect in this group of direct causes. The positive difference between prediction and observation could be explained by possible difficulties on reactivation of the health system and routine obstetric controls associated with a decrease in the MMR [5, 6, 8]. Nevertheless, from 2020–2021, both the public and private health systems of Chile implemented coordinated and monitored daily health strategies focused on guaranteeing quality care for pregnant women in situations of greater vulnerability, trying to ensuring prenatal check-ups and postnatal care, and access to emergency obstetric services.

On the other hand, both analyses identified groups of indirect causes, particularly non-respiratory causes in Chile for 2020 and 2021. This was consistent with other studies [21, 23–25]. Indirect causes of maternal death are linked to medical conditions that are not directly related to the pregnancy. These conditions often include non-transmissible comorbidities, such as diabetes, asthma, and obesity. Notably, COVID-19 has disproportionately affected groups with these preexisting conditions [21, 23–25]. Other studies reported indirect respiratory causes as a relevant group in contributing to maternal mortality. Acute respiratory distress syndrome and pneumonia associated with SARS-CoV-2 have been indicated as leading causes of death among pregnant women [21]. The Mexican study reported 32% increase in the group of MMRs due to indirect respiratory causes compared to the previous period (2011–2019) [15]. In regard to previous studies on the effect of pandemic events, such as the case of Argentina for emergent H1N1 in 2009, a specific increase in maternal mortality due to respiratory causes was observed [14]. In contrast, our study identified only a slight increase in indirect respiratory causes. One hypothesis that could explain these differences would be the criterion for coding the causes of maternal deaths.

In Chile, following WHO suggestions, maternal deaths aggravated by SARS-CoV-2 are assigned to code O98.5 (non-respiratory infectious indirect) accompanied by code U07.1 or U07.2, depending on confirmation of the presence or absence of the virus [12]. In this sense, in 2021, the group of causes of maternal deaths that had the greatest increase compared to 2020 was the group that included the deaths assigned to these codes. This increase in maternal mortality from SARS-CoV-2 in Chile in 2021 compared to 2020 coincides with the findings of a study carried out in Brazil, where it was observed that, in 2021, maternal mortality increased with the arrival of the Gamma variant [26].

Physiological changes during pregnancy may favour complications in the presence of SARS-CoV-2 [27]. The change from cellular to humoral immunity during pregnancy is related to increased susceptibility to viral infections in pregnant women [28]. At present, however, the information available on the increased susceptibility to SARS-CoV-2 during pregnancy is contradictory. In fact, it has been proposed that pregnant women, in relation to the general population, may have a lower morbidity and mortality [21]. In this sense, it has been suggested that human chorionic gonadotropin and progesterone lower the proinflammatory activity of TH-1, decreasing tumour necrosis factor [21]. This modulating effect is hypothesized to protect against the cytokine storm and, therefore, mortality associated with SARS-CoV-2 during pregnancy [29]. However, the burden of pre-existing chronic diseases could nullify this hypothetical protective factor during pregnancy. Thus, according to the findings of a study in Brazil, pregnancy and postpartum may be important risk factors associated with severe COVID-19 [26, 30].

A fresh perspective in this discussion, considering that the MMR from direct obstetric causes has increased, suggests that the pathological pathways of this emerging coronavirus have negative effects on pregnancy itself. Another hypothesis suggests that this novel

coronavirus predominantly affects the MMR from indirect obstetric causes that are frequently associated with pre-existing conditions such as diabetes and obesity. However, the absence of detailed information on underlying causes hinders the differentiation between pregnant/post-partum with comorbidities or chronic diseases. To advance, further research is essential for a comprehensive understanding of these complex interactions.

The strengths of this natural experiment include the use of two robust statistical analysis techniques, ITS and prediction with ARIMA models. The first allowed us to determine the causal effect of a specific factor by controlling for the temporal trend before and after the event of interest. Second, we aimed to forecast and estimate the MMR in the absence of the COVID-19 pandemic with good accuracy levels. Similarly, population-based studies reduce the different sampling biases typical of observational studies, under the condition that the entire population of pregnant women exposed or not exposed to the SARS-CoV-2 is included, thus covering all deaths potentially attributable to it. In addition, according to our knowledge, this is the first Chilean study to consider at the national level the impact of the pandemic on maternal mortality and to distinguish groups of causes of mortality with the ICD-10 classification. This provides relevant epidemiological information since population-based estimates of maternal mortality in Latin America during the pandemic period are insufficient [31].

On the other hand, among the limitations is that the data on the MMR in Chile, currently published by the Ministry of Health for the years 2020–2021, are provisional and susceptible to further adjustments. Additionally, it may be subject to underreporting or misclassification of COVID-19 cases. In interpreting our results based on annual data, we emphasize the need for caution. While our findings provide valuable insights, fine-grained analyses—such as exploring shorter time intervals or regional levels—would further enhance the robustness of our evidence. However, given the limited number of cases registered monthly, this typology of analysis is more robust when conducted with annual national aggregates data. Furthermore, the maternal mortality information is published annually. In addition, the level of certainty of prediction values may have been affected by the absence of cases of death for some years. However, two error measures, the mean absolute error and mean absolute percentage error, were estimated to give a measure of a good level of accuracy. Likewise, multiple factors were not controlled for, including the number of pregnant women who received SARS-CoV-2 vaccines and the number of beds and respirators available during this period, so the results should be interpreted with caution.

In conclusion, the Chilean maternal mortality registry was revealed as a useful tool to evaluate the specific effects of SARS-CoV-2 on the health of pregnant women. In Chile, from 2020–2021, there was an increase in total maternal mortality associated with SARS-CoV-2. In particular, the pandemic contributed to an increase in the MMR due to indirect causes, specifically, non-respiratory and infectious causes. Similarly, there was an increase in the MMR due to direct obstetric causes. This underscores the critical need for robust and adaptable maternal healthcare systems during worldwide health emergencies. It is imperative, then, to prevent damage to maternal and child health during epidemic periods and that health strategies be implemented that focus on ensuring continuity of quality care, assuring recommended prenatal and postnatal check-ups and access to specialized obstetric services for high-risk pregnancies.

## Supporting information

**S1 Table. Characteristics of the ARIMA models, selection criteria and accuracy indicators of the prediction of the maternal mortality ratio (MMR) groups of mortality, in Chile (1997–2021).**
(DOCX)

**S1 Data. Database Chile MMR 1997–2021.**
(XLSX)

**S1 File. Alternative language abstract.**
(DOCX)

## Author Contributions

**Conceptualization:** Yordanis Enriquez, María Elena Critto, Elard Koch.

**Data curation:** María Elena Critto.

**Formal analysis:** Yordanis Enriquez, María Elena Critto, Elard Koch.

**Methodology:** Yordanis Enriquez, Elard Koch.

**Project administration:** María Elena Critto, Elard Koch.

**Supervision:** Elard Koch.

**Validation:** Elard Koch.

**Writing – original draft:** Yordanis Enriquez, María Elena Critto.

**Writing – review & editing:** Yordanis Enriquez, María Elena Critto, Ruth Weinberg, Lenin de Janon Quevedo, Aliro Galleguillos, Elard Koch.

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
