## [Decision Letter · Decision Letter 0]

4 Apr 2024

PGPH-D-24-00068

Title - Impact of emerging SARS-CoV-2 on total and cause-specific maternal mortality: A natural experiment in Chile during the peak of the outbreak SARS-CoV-2 Impact on Maternal Mortality: Chile’s Outbreak Study

Dear Dr. Koch,

Thank you for submitting your manuscript to PLOS Global Public Health. After careful consideration, we feel that it has merit but does not fully meet PLOS Global Public Health’s publication criteria as it currently stands. Therefore, we invite you to submit a revised version of the manuscript that addresses the points raised during the review process.

We look forward to receiving your revised manuscript.

Kind regards,

Edina Amponsah-Dacosta, Ph.D., MPH

Academic Editor

Journal Requirements:

1. Please amend the title to remove "Title"

2. We do not publish any copyright or trademark symbols that usually accompany proprietary names, eg  ©, ®, ™  (e.g. next to drug or reagent names). Please remove all instances of trademark/copyright symbols throughout the text, including  ® on page 7.

3. In the online submission form, you indicated that "The data underlying this article is available from Departamento de Estadísticas de Información de Salud del Gobierno de Chile, https://deis.minsal.cl/. The derived data generated in this research will be shared on reasonable request to the corresponding author". All PLOS journals now require all data underlying the findings described in their manuscript to be freely available to other researchers, either 1. In a public repository, 2. Within the manuscript itself, or 3. Uploaded as supplementary information.

Additional Editor Comments (if provided):

Reviewers' comments:

Reviewer's Responses to Questions

**Comments to the Author**

1. Does this manuscript meet PLOS Global Public Health’s publication criteria? Is the manuscript technically sound, and do the data support the conclusions? The manuscript must describe methodologically and ethically rigorous research with conclusions that are appropriately drawn based on the data presented.

Reviewer #1: Yes

Reviewer #2: Yes

2. Has the statistical analysis been performed appropriately and rigorously?

Reviewer #1: Yes

Reviewer #2: Yes

3. Have the authors made all data underlying the findings in their manuscript fully available (please refer to the Data Availability Statement at the start of the manuscript PDF file)?

Reviewer #1: Yes

Reviewer #2: Yes

4. Is the manuscript presented in an intelligible fashion and written in standard English?

Reviewer #1: Yes

Reviewer #2: Yes

5. Review Comments to the Author

Reviewer #1: Dear Authors,

I appreciate the opportunity to review this important manuscript. Describing and analyzing the potential impacts of the Covid-19 pandemic on maternal mortality is of utmost importance for the enhancement of public health policies and better preparation for future epidemics.

Below, I present some comments and suggestions aimed at contributing to the improvement of your study.

• Standardization of Terminologies: Throughout the manuscript, the authors use terms such as ‘SARS-CoV-2’, ‘emerging SARS-CoV-2’, ‘SARS-CoV-2 pandemic’, and ‘SARS-CoV-2 outbreak’. In some paragraphs, it is unclear whether the authors are specifically referring to cases of infection by the SARS-CoV-2 virus in pregnant and postpartum women, or to the scenario of the COVID-19 pandemic (which involved a set of factors that may have influenced maternal mortality, e.g., infections by SARS-CoV-2 virus, potential restrictions on access to healthcare services, disruption of prenatal consultation routines; inequalities in access to vaccination; resources and health policies employed in addressing the health crisis, among others). Therefore, I suggest that the authors standardize the terminologies, using SARS-CoV-2 to refer to the virus, and COVID-19 pandemic to refer to the period of the Public Health Emergency of International Concern due to SARS-CoV-2.

• Title: I also recommend including the period in the title: “Impact of (…) in Chile during the peak of the SARS-CoV-2 outbreak, 2020-2021”

• Abstract: For better alignment with the objective described in the main text of the manuscript, I suggest describing the background as “This study estimated the effects of the COVID-19 pandemic on maternal mortality in Chile between 2020 and 2021.”

• Introduction: In the last paragraph of introduction, it is suggested to omit the sentence "To achieve this, we employed a time-series design exploiting long-term annual trend information and used ARIMA models to forecast expected mortality under the hypothesis that previous mortality trends would continue in the absence of the pandemic virus-related mortality burden". These details should be allocated in the methods section.

• Methods: a) Please, present a descriptive table showing the number and percentage of maternal deaths in Chile, according to the groups of causes and the year of occurrence. If possible, also specify the number and percentage of maternal deaths due to COVID-19 in 2020 and 2021. This would be useful for viewing the distribution of these deaths before and during the COVID-19 pandemic.

b) Did the authors work with the underlying cause of maternal death? The authors need to specify the selection criteria adopted when, in a single maternal death notification, there were ICD-10 codes belonging to both direct and indirect obstetric cause groups. Similarly, it is necessary to clarify the procedure followed when the codes O98.5 and O99.5 appeared in the same maternal death notification. There is an example on page 12 of the document "International Guidelines for Certification and Classification (Coding) of COVID-19 as Cause of Death" <https: encurtador.com.br="" ilntz="">.

c) The authors applied three statistical methods for time-series analysis: linear logarithmic regression (Joinpoint), interrupted time series (ITS), and autoregressive integrated moving average (ARIMA). Although the three methods together provide more information, it has resulted in an extended manuscript. A suggestion for the authors is to leverage the Joinpoint analysis to produce another manuscript, and in this one, to prioritize ITS and ARIMA analyses, thereby making the text more concise and focused on the objective of assessing the impact of the COVID-19 pandemic on general maternal mortality and by subgroups of causes.

d) The statistical analyses are well described and appropriate. However, in Figure 2, it can be seen that, in some time series, the regression line for the pre-interruption period does not seem to fit the points so well. I wonder if the authors tested analyzing different sizes of time series (different temporal intervals) and/or statistical methods for ITS, to assess the robustness of the results.

Turner, S.L., Karahalios, A., Forbes, A.B. et al. Comparison of six statistical methods for interrupted time series studies: empirical evaluation of 190 published series. BMC Med Res Methodol 21, 134 (2021). https://doi.org/10.1186/s12874-021-01306-w

Results and Discussion: a) The results and discussion sections are well-written. However, I have a concern regarding the following interpretations in the discussion:

"Our findings suggest that SARS-CoV-2 predominantly impacts women with pre-existing conditions and comorbidities, such as diabetes and obesity." (Lines 382-383)

"This implies that the increase in maternal mortality primarily occurred among women with pre-existing comorbidities." (Lines 412-413)

It is unclear how the authors arrived at this conclusion, given that the study's data were aggregated, and it probably was not possible to differentiate whether the pregnant/postpartum women who died from COVID-19 had comorbidities like diabetes, obesity, etc. COVID-19 can lead to maternal death, even in the absence of comorbidities. Furthermore, ARIMA and ITS analyses indicate an increase in maternal deaths mainly due to indirect infectious causes, while maternal deaths from indirect non-infectious causes remained within the forecasts for 2020 and 2021.

b) It is recommended to comment on the coverage and quality of data; about the possibility of underreporting of maternal deaths in the country; the possibility that diagnoses from the group of indirect respiratory causes (pneumonia; acute respiratory distress syndrome) might have been cases of undiagnosed Covid-19.

Once again, I appreciate the opportunity, and I hope my remarks will be useful for the refinement of this study. I emphasize the relevance of the topic and wish success to the authors.</https:>

Reviewer #2: This an interesting and well written study on an important problem.

I was suprised by the lack of impact of the crisis on obstetrical causes of maternal mortality for which i assumed that good pregnancy follow-up had a beneficial effect.

the conclusion is a bit long perhaps general considerations should be moved to the discussion to keep a short conclusion with a crisp message on what was the hypothesis and what was found and the concordance of the 2 methods.

The analysis is complex and has several components so the message is not so easy to wrap one's head around.

6. PLOS authors have the option to publish the peer review history of their article (what does this mean?). If published, this will include your full peer review and any attached files.

**Do you want your identity to be public for this peer review?** For information about this choice, including consent withdrawal, please see our Privacy Policy.

Reviewer #1: No

Reviewer #2: No

---

## [Decision Letter · Decision Letter 1]

21 Jun 2024

Effects of emerging SARS-CoV-2 on total and cause-specific maternal mortality: A natural experiment in Chile during the peak of the outbreak, 2020-2021

PGPH-D-24-00068R1

Dear Dr. Koch,

We are pleased to inform you that your manuscript 'Effects of emerging SARS-CoV-2 on total and cause-specific maternal mortality: A natural experiment in Chile during the peak of the outbreak, 2020-2021' has been provisionally accepted for publication in PLOS Global Public Health. **It is important that you take note of the final comments raised by Reviewer 1 with regards to the Abstract and Table 1**.

Best regards,

Edina Amponsah-Dacosta, Ph.D., MPH

Academic Editor

Reviewer Comments (if any, and for reference):

Reviewer's Responses to Questions

**Comments to the Author**

1. If the authors have adequately addressed your comments raised in a previous round of review and you feel that this manuscript is now acceptable for publication, you may indicate that here to bypass the “Comments to the Author” section, enter your conflict of interest statement in the “Confidential to Editor” section, and submit your "Accept" recommendation.

Reviewer #1: All comments have been addressed

Reviewer #2: All comments have been addressed

2. Does this manuscript meet PLOS Global Public Health’s publication criteria? Is the manuscript technically sound, and do the data support the conclusions? The manuscript must describe methodologically and ethically rigorous research with conclusions that are appropriately drawn based on the data presented.

Reviewer #1: Yes

Reviewer #2: Yes

3. Has the statistical analysis been performed appropriately and rigorously?

Reviewer #1: Yes

Reviewer #2: Yes

4. Have the authors made all data underlying the findings in their manuscript fully available (please refer to the Data Availability Statement at the start of the manuscript PDF file)?

Reviewer #1: Yes

Reviewer #2: Yes

5. Is the manuscript presented in an intelligible fashion and written in standard English?

Reviewer #1: Yes

Reviewer #2: Yes

6. Review Comments to the Author

Reviewer #1: Dear authors,

The manuscript "Effects of emerging SARS-CoV-2 on total and cause-specific maternal mortality: A natural experiment in Chile during the peak of the outbreak, 2020-2021" is well-written and well-developed. The authors have satisfactorily addressed the recommendations from the first review session. My suggestion is that this article be accepted for publication.

Nonetheless, I would like to make an additional recommendation regarding the abstract. There is nothing incorrect in this section of the article, but I believe it is important to add a concise sentence stating that "in Chile, following WHO suggestions, maternal deaths aggravated by SARS-CoV-2 are assigned to code O98.5 (non-respiratory infectious indirect) accompanied by code U07.1 or U07.2, depending on confirmation of the presence or absence of the virus", as the authors made clear in the main text.

This recommendation is due to the fact that COVID-19 is an infectious systemic disease, but its clinical expression is primarily respiratory. By stating that the increase in maternal mortality occurred more intensely in the non-respiratory causes group (with infectious causes, including maternal deaths from COVID-19, being included in this group), it may confuse readers who are less familiar with the ICD-10 classification codes. Therefore, it is necessary to include this explanation in the abstract.

Another observation is that in Table 2, the delta value for total MMR is incorrect. Please review this value and the table's footnotes.

Finally, I congratulate the authors on their work and reiterate my best wishes for health and peace.

Sincerely,

Anonymous Reviewer 1

Reviewer #2: the authors have followed my suggestion

i have no further observations

7. PLOS authors have the option to publish the peer review history of their article (what does this mean?). If published, this will include your full peer review and any attached files.

**Do you want your identity to be public for this peer review?** For information about this choice, including consent withdrawal, please see our Privacy Policy.

Reviewer #1: No

Reviewer #2: No
